# The Role of Muscle Decline in Type 2 Diabetes Development: A 5-Year Prospective Observational Cohort Study

**DOI:** 10.3390/nu11040834

**Published:** 2019-04-12

**Authors:** Katarzyna Maliszewska, Edyta Adamska-Patruno, Joanna Goscik, Danuta Lipinska, Anna Citko, Aleksandra Krahel, Katarzyna Miniewska, Joanna Fiedorczuk, Monika Moroz, Maria Gorska, Adam Kretowski

**Affiliations:** 1Department of Endocrinology, Diabetology and Internal Medicine, Medical University of Bialystok, Poland M.C. Skłodowskiej-Curie 24A, 15-276 Bialystok, Poland; danuta.lipinska@umb.edu.pl (D.L.); ola.swiderska123@gmail.com (A.K.); mgorska25@wp.pl (M.G.); adamkretowski@wp.pl (A.K.); 2Clinical Research Centre, Medical University of Bialystok, Poland; M.C. Skłodowskiej-Curie 24A, 15-276 Bialystok, Poland; edyta.adamska@wp.pl (E.A.-P.); joanna.goscik@umb.edu.pl (J.G.); annacitko@gmail.com (A.C.); katarzyna.miniewska@gmail.com (K.M.); j.fiedorczuk@wp.pl (J.F.); monika_bakun@wp.pl (M.M.)

**Keywords:** Type 2 diabetes, muscle decline, observational cohort study, insulin resistance

## Abstract

The major risk factors of T2DM (type 2 diabetes mellitus) development are still under investigation. We evaluate the possible risk factors associated with type 2 diabetes (T2DM) in adult subjects during a five-year prospective cohort study. We recruited 1160 subjects who underwent oral glucose tolerance test, anthropometric measurements, and body composition and body fat distribution analysis at a baseline visit and again at follow-up after approximately five years. The conclusions of this study are based on observation of 219 subjects who attended both the first and follow-up visits. The fasting serum insulin was measured, and HOMA-IR (homeostatic model assessment of insulin resistance) was calculated. During the follow-up period, T2DM was diagnosed in 7.4% of participants, impaired fasting glucose in 37.7%, and impaired glucose tolerance in 9.3%. Logistic regression models, adjusted for age, were constructed. The changes in glucose concentration, visceral fat tissue content, insulin resistance, and %loss of muscle mass were chosen as the potential predictors for T2DM development. A set of independent variables was extracted. The constructed feature set comprised change in HOMA-IR (OR (odds ratio) = 1.01, *p* < 0.01) and change in %loss of muscle mass (OR = 0.84, *p* < 0.03). With an aim to validate the prediction capability using the selected attributes, a support vector machine classifier and leave-one-out cross-validation procedure was applied, yielding 92.78% classification accuracy. Our results show the correlation between the %loss of muscle mass and T2DM development in adults, independent of changes in insulin resistance.

## 1. Introduction

Type 2 diabetes (T2DM) and cardiovascular disease are the leading causes of death and illness in developing societies. Moreover, they are becoming the dominating health problems worldwide [1]. The major risk factors of T2DM development are still under investigation. Muscle mass is the primary tissue contributing to whole-body insulin-mediated glucose disposal, and its decline contributes to insulin resistance, which plays a crucial role in T2DM pathogenesis. The muscles are responsible for more than two-thirds of excess glucose disposal after a meal intake and for nearly all non-oxidative glucose storage, as glycogen, under hyperinsulinemia conditions [2].

Outcomes from the Third National Health and Nutrition Examination Survey indicate that muscle mass is inversely associated with insulin resistance and prediabetes states [3]. Age-associated loss of muscle mass begins in the fifth decade of life, and up to 50% of muscle may be lost by the age of 90 years [4]. Sarcopenia, independent of obesity, is associated with adverse glucose metabolism. This association is strongest in individuals under 60 years of age, which suggests that low muscle mass may be an early predictor of type 2 diabetes susceptibility [5]. Findings from a representative cohort of Australian men aged 35 to 81 years showed that muscle mass and strength are strong protective factors, independent of insulin resistance and abdominal fat accumulation [6]. Interventional studies confirmed that lifestyle modification reduced the risk of T2DM by 58%, whereas treatment with metformin reduced the risk of diabetes only by 31% [7], and there is strong evidence that physical activity protects against T2DM development in subjects with insulin resistance [2]. The aim of our study was to evaluate the possible risk factors associated with T2DM development in adult subjects during a five-year prospective observational cohort study.

## 2. Materials and Methods

### 2.1. Ethics Statement

Written informed consent was obtained from all participants before inclusion in the study. The study protocol was approved by the local Ethics Committee of the Medical University of Bialystok, Poland (R-I-002/35/2009 and R-I-002/32/2014). The study methods were performed in accordance with the ethical standards on human experimentation and with the Helsinki Declaration of 1975 as revised in 1983.

### 2.2. Subjects and Study Design

The study group comprised 1160 Polish, Caucasian volunteers (583 women, 568 men; 18–65 years old; mean age 44 years old) without a known history of dysglycemia. All study participants were from the Podlasie region and were recruited for the 1000PLUS cohort, as a part of our larger project [8,9,10,11,12,13].

Among the study population, 763 subjects were overweight/obese and 356 had a body mass index (BMI) of <25 kg/m^2^. In 2015–2016, 219 participants who were free from impaired fasting glucose (IFG) and impaired glucose tolerance (IGT) at the baseline visit underwent a five-year follow-up examination.

### 2.3. Anthropometric and Body Composition Measurements

During both visits, anthropometric data were recorded. The weight, height, and waist and hip circumferences were measured, and the BMI and waist/hip ratio (WHR) were calculated.

Body composition analysis was performed using the multi-frequency bioimpedance method (Maltron BioScan 920-2, Maltron International Ltd., UK), and skeletal muscle mass, total body fat, visceral adipose tissue (VAT), subcutaneous adipose tissue (SAT), and the VAT/SAT ratio were estimated.

### 2.4. Blood Collection and Biochemical Analysis

Non-diabetic subjects underwent an oral glucose tolerance test (OGTT) according to the World Health Organization (WHO) recommendations [14] with a 75 g oral glucose dose. The participants were instructed to fast for 8–12 h prior to the test and to not restrict carbohydrate intake in the three days prior to the test. Blood was collected at 0, 30, 60, and 120 min during the OGTT.

The plasma glucose concentration was analyzed by the enzymatic method with hexokinase (Roche Diagnostics International Ltd., Switzerland). Serum insulin concentrations were evaluated in duplicate by the immunoradiometric assay (IRMA, Diasource ImmunoAssay S.A., Nivelles, Belgium).

### 2.5. Calculations

The diagnoses of T2DM and prediabetes were made based on WHO criteria [14]. IFG (impaired fasting glucose) was defined as fasting plasma glucose between 0.1 mmol/L and 6.9 mmol/L (100 mg/dl to 125 mg/dl). IGT (impaired glucose tolerance) was defined by an elevated 2 h plasma glucose concentration (≥140 and <200 mg/dl or 7.8 mmol/L to 11.1 mmol/L) after a 75 g glucose load on the oral glucose tolerance test (OGTT) in the presence of the fasting plasma glucose concentration of <126 mg/dl (7.0 mmol/L). BMI was calculated as weight in kilograms (kg) divided by height in meters (m) squared (kg/m^2^), and the WHR was calculated by dividing waist circumference (in cm) by hip circumference (cm). To evaluate insulin resistance, we used the homeostasis model assessment: HOMA-IR = fasting insulin (µU/mL) × fasting glucose (mmol/L)/22.5; and β-cell function: HOMA-B = [insulin (µU/mL) × 20]/[glucose (mmol/L) − 3.5] [15]. The corrected insulin response at 30 min of the OGTT (CIR 30 min) was calculated according to Sluiter [16] as ([serum insulin_30min_ (pmol/l)/6.45] × 100)/(plasma glucose_30min_(mmol) × [plasma glucose_30min_ (mmol/L) − 3.89]) to evaluate the early phase of insulin secretion. The percentage change in muscle mass was evaluated using the formula (muscle mass in % on second visit) divided by (muscle mass in % on first visit) multiplied by 100%.

### 2.6. Statistical Analysis

Generalized linear models with the logit link function and adjusted for age at the second visit were constructed to determine a set of features enabling the prediction of T2DM occurrence under the assumption that T2DM was not observed at the first visit. Among other factors, changes in glucose concentration, visceral fat tissue volume, HOMA-IR, %loss of muscle mass, BMI, and VAT/SAT ratio were chosen as potential predictors of T2DM and were included in the models (a total of 36 potential risk factors were identified: BMI change, BMI percentage change, WHR (waist-hip ratio) change, WHR percentage change, waist circumference percentage change, waist circumference change, hip circumference percentage change, hip circumference change, percentage change of fasting glucose, change of fasting glucose, HOMA-IR percentage change, HOMA-IR change, HOMA-B (homeostatic model assessment B) percentage change, HOMA-B change, HbA1c percentage change, HbA1c change, change of fat mass (%), percentage change of fat mass (%), percentage change of muscle mass (kg), change of muscle mass (kg), percentage change of VAT (cm³), change of SAT (cm³), percentage change of SAT (%), change in % loss of muscle mass (%), percentage change of % loss of muscle mass (%), change of VAT (%), percentage change of VAT (%), percentage change of SAT (cm³), change of SAT (%), change of VAT/SAT ratio, percentage change of VAT/SAT ratio, change of glucose (in 2 h of OGTT—oral glucose tolerance test), percentage change of glucose (in 2 h of OGTT), change of fat mass (kg), percentage change of fat mass (%), and change of VAT (cm³)).

The odds ratios reported in Tables 3 and 4 come directly from the models. The features that showed a statically significant impact on T2DM development were investigated further. As it was highly probable that some of the selected features were somehow dependent, that is, redundant, a set of independent attributes was extracted based on the determined threshold value of the correlation coefficient. As the joint probability distributions were not approximately normal, Spearman’s correlation coefficient [17] was calculated for each pair of previously selected variables. Correlations were considered as statistically significant only in situations when the corresponding FDR (false discovery rate)-adjusted [18] *p*-value was less than 0.05. The *p*-value adjustment procedure was applied because of the multiple testing phenomenon. The greatest statistically insignificant and smallest statistically significant correlation coefficients were used to set the threshold value for the correlation coefficient—the association between two variables was considered not significant if their correlation coefficient was lower than the determined threshold. Using the described methodology, a set of independent features was constructed. With an aim to validate the T2DM prediction capability using the selected attributes, a support vector machine classifier [19] and leave-one-out cross-validation procedure was applied. All calculations were carried out with the use of the R software environment [20] and its packages—caret [21] for feature selection, e1071 [22] for classification and validation, and R’s base packages.

## 3. Results

The characteristics of the studied population at the baseline and follow-up visits are presented in Table 1.

During the follow-up visit, T2DM was diagnosed in 7.4% (*N* = 16) of participants, impaired fasting glycose (IFG) in 37.7% (*N* = 78), and impaired glucose tolerance (IGT) in 9.3% (*N* = 19); 93 subjects (45.6%) remained normoglycemic (Table 2).

The strongest association with T2DM susceptibility was found for age (*p* < 0.01). Increases in waist and hip circumference showed a statistically significant impact on IFG development (OR = 1.06, *p* < 0.01; OR = 1.05, *p* = 0.03, respectively) (Table 3) after eliminating the influence of subject age as a confounder.

When considering subjects diagnosed with IGT during the follow-up visit (without an IGT state at the first visit), we found increases in hip circumference (OR = 1.09, *p* = 0.04), visceral fat accumulation (OR = 1.03, *p* = 0.01), and VAT/SAT ratio (OR = 1.02, *p* = 0.02), and a decrease in subcutaneous fat accumulation (OR = 0.92, *p* = 0.01). All of these associations were found to be statistically significant after taking the defined confounder into consideration (Table 4).

Focusing on the aim of the study, that is, the determination of potential risk factors for T2DM development, a set of 36 features likely related with T2DM development were identified. The set comprised, among other factors, changes in BMI, glucose and insulin concentrations (measured at different time points during the OGTT), WHR, HOMA-IR, VAT and SAT tissue, and %loss of skeletal muscle mass. After applying a procedure of feature selection (on features showing a statistically significant impact on T2DM development), as described in the Statistical Analysis section, two independent sets of features were discovered. The first set comprised changes in HOMA-IR, while the second set comprised changes in %loss of muscle mass tissue measured in various ways. The percentage change in HOMA-IR level and percentage change in %loss of muscle mass (in terms of percentage of total body mass) assured the best classification accuracy, equal to 92.78%. Additionally, descriptive statistics and related odds ratios for selected features are presented (Table 5). The data show that a reduction of %loss of muscle mass increases the risk of T2DM development (OR = 0.84, *p* = 0.02). Also, an increase in insulin resistance (estimated by HOMA-IR calculation) (OR = 1.01, *p* < 0.01) is a significant risk factor for glucose dysmetabolism development.

## 4. Discussion

Our study showed that a reduction of muscle mass over a five-year time span is a risk factor for T2DM susceptibility, independent of insulin resistance, in adult subjects. Additionally, we showed that the development of prediabetes and T2DM was associated with changes in some other risk factors. IFG was statistically significant in relation to changes in waist and hip circumference, while increases in hip circumference and VAT accumulation determined the occurrence of IGT. T2DM development, in contrast, was associated with a %loss of muscle mass and an increase in insulin resistance. Moreover, both risk factors together result in an exact risk of over 90% for T2DM susceptibility.

Our findings are consistent with results from the Korean Genome Epidemiology Study, a large 12-year prospective cohort study, in which it was shown that low muscle mass was strongly associated with an increased risk of T2DM, independently of obesity, in middle-aged Korean adults during a nine-year follow-up period [23]. Also, other authors reported that type 2 diabetes was associated with reduction of muscle strength and with an excessive decline of skeletal muscle mass over a six-year follow-up period, based on 2675 subjects, in the Health, Aging, and Body Composition Study [24]. The results from a large national study carried out in 1195 Australian men suggested that muscle mass and strength were inversely associated with metabolic syndrome regardless of insulin resistance and accumulation of abdominal fat [6]. Consistent with those findings are the results from Srikanthan P et al. who found that relative muscle mass is inversely associated with prediabetes states and lower insulin sensitivity. The same author showed that sarcopenia in subjects under 60 years old was strongly associated with more significant insulin resistance and a higher prevalence of T2DM in the obese population [3,5]. Analyzing the above-mentioned data, we should underline that lost skeletal muscle mass was defined as a one-time measurement, not as it was performed in our study—an observational change in muscle mass over years.

An explanation for the reduced muscle mass among middle-aged subjects and the increased risk of T2DM could be the different types of muscle atrophy. Type II muscle fibers, which are less dependent on metabolic insulin action [25], are reduced more than type I fibers in age-related muscle atrophy [26]. The decreased size of type II fibers leads to reduced mitochondrial activity and to insulin resistance [27]. In summary, it appears that IR, loss of muscle mass, and change in the type of muscle fibers independently or additively alter glucose homeostasis with aging. In aged subjects, the relations between insulin and skeletal muscle decline are based on muscle protein resistance to anabolic insulin action and on defects associated with insulin-induced vasodilation and mammalian target of rapamycin signaling (mTORC1) [27].

Moreover, the ageing of muscle is characterized by fat infiltration [28], which occurs at a macroscopic level between muscle groups and at a microscopic level between and inside myocytes. Intramyocellular lipid accumulation may be related to reduced oxidative mitochondrial capacity. Greater ectopic lipid content within skeletal muscle in obese elderly patients may be an explanation for dysglycemia and diabetes in obese sarcopenic adults [29]. Additionally, a mechanistic link between expansion of visceral fat tissue and muscle atrophy has been suggested to decrease the expression of contractile proteins in human myotubes co-cultured with visceral adipocytes from obese subjects [30].

Moreover, Srikanthan P et al. [2] showed that in young and middle-aged individuals, sarcopenia is associated with higher levels of serum CRP (C reactive protein). Insulin resistance and myostatin/activin A activation most probably contribute to the wasting of muscles; the presence of insulin resistance (as a type of chronic inflammation) leads to autophagy, degradation of muscle proteins (i.e., via the ubiquitin–proteasome proteolytic pathway) [31], and mitochondrial dysfunction. Some studies have shown that in obesity or T2DM, the size of muscle mitochondria is smaller and they have abnormalities in their internal membranes (with the presence of vacuoles) when compared with those in healthy and normal-weight subjects. A low mitochondrial size correlates with low glucose disposal rate and insulin sensitivity [32,33]. These processes ultimately lead to loss of muscle mass or muscle strength, or both. The increasing severity of insulin resistance stimulates pathways resulting in accelerated loss of muscle, and the cycle repeats itself [34].

Reduced insulin sensitivity in skeletal muscle is the main defect that links obesity and type 2 diabetes. Insulin resistance and suppressed IGF-1 signaling drive the loss of muscle mass. One of the potential mechanisms mediating the development of muscle mass decline and insulin resistance is reduced expression of the E3 ubiqitin enzymes, which participate in muscle protein degradation, a mechanism not observed in age-related sarcopenia [27]. As a result of skeletal muscle loss, a 30% lower basal metabolic rate is observed in people between 20 and 70 years old [35]; meanwhile, caloric intake does not necessarily decrease over the lifespan [36] and may lead to increased accumulation of fat adipose tissue. Fat tissue accumulation, as mentioned above, can cause muscle mass decline, along with other factors contributing to age-related muscle and strength loss, with physical inactivity being probably the most important factor. Data from interventional studies showed that lifestyle modification reduces the risk of T2DM by 58% [37], which is strong evidence that exercise protects against T2DM development in patients with insulin resistance. Moreover, results from post-hoc analyses of the Diabetes Prevention Program (DPP) and its outcomes study indicate that physical activity over the entire DPP was inversely related to new-onset diabetes [38]. Smith et al., using the cubic spline model, observed a 26% decrease in T2DM risk among subjects who achieved 11.25 MET (metabolic equivalent) h/week (equivalent to 150 min/week of moderate activity) [39].

Our study also presents some limitations, one of them being the number of subjects who attended a follow-up visit, and we plan to work on a better participation response in the next follow-up visits. Second, we used the bioimpedance method to estimate muscle mass, and this measurement relies on the relationship between body composition and body water content, which may be disturbed in pathological states. Therefore, to minimize the impact of other conditions, subjects with severe heart failure and renal insufficiency were not included in the analysis. To reduce the possibility of measurement errors, participants fasted before the bioelectrical impedance analysis (BIA) assessment, and their hydration status was carefully monitored. This is a single-center study performed in Poland. Therefore, external validation in other countries or races is necessary to confirm the findings of this study. In summary, our five-year-long prospective cohort study, based on a large number of samples obtained from the general Polish population, showed that the %loss of muscle mass correlated significantly with prediabetes and T2DM development, independent of insulin resistance. This underlines the problem, especially among young adults who are at high risk of metabolic diseases and additionally have limited physical activity. Moreover, our results highlight the importance of monitoring muscle mass in assessing an individual’s metabolic health and suggest the potential role of physical activity in metabolic disorder prevention. It would be reasonable to conduct interventional studies to evaluate the effectiveness of exercise interventions to improve muscle mass and glucose metabolism in terms of the risk of T2DM.

## Figures and Tables

**Table 1 nutrients-11-00834-t001:** Characteristics of the studied population.

	Visit I	Visit II *
N	1160	219
Women/Men	583/568 (51%/49%)	103/116 (47%/53%)
Mean age (years)	44	49
BMI > 25 kg/m2	763 (70%)	141 (71%)
BMI < 25 kg/m2	356 (30%)	59 (29%)
IFG (N)	298	78
IGT (N)	88	19
T2DM(N)	96	16

* for visit II, 219 participants who were free from impaired fasting glucose (IFG) and impaired glucose tolerance (IGT) at the baseline visit were enrolled. BMI—body mass index; T2DM—type 2 diabetes mellitus.

**Table 2 nutrients-11-00834-t002:** Characteristics of subjects with prediabetes and T2DM, diagnosed at the follow-up visit.

	Healthy	IFG	IGT	T2DM
N/%	93/45.6%	78/37.7%	19/9.3%	16/7.4%
Age [years]	42 ± 1.61	49 ± 1.55 (*p* < 0.03)	56 ± 2.20 (*p* < 0.01)	56 ± 3.03 (*p* < 0.01)
BMI [kg/m²]	25.70 ± 0.54	29.51 ± 0.84 *p* < 0.01	30.73 ± 1.38 *p* = 0.03	34.46 ± 2.59 *p* = 0.02
Waist circumference [cm]	88.21 ± 1.52	100.42 ± 1.94 *p* < 0.01	103.33 ± 3.43 *p* = 0.02	113.53 ± 6.27 *p* < 0.01
Hip circumference [cm]	95.67 ± 1.12	104.75 ± 1.49 *p* < 0.01	109.11 ± 2.54 *p* < 0.01	114.60 ± 4.89 *p* < 0.01
Muscle mass [%]	35.35 ± 0.61	31.78 ± 0.69 *p* = 0.02	28.61 ± 0.99 *p* < 0.01	28.74 ± 2.03 *p* = 0.03
Subcutaneous fat [%]	76.12 ± 0.74	70.21 ± 0.89 *p* < 0.01	68.83 ± 1.95 *p* = 0.06	64.25 ± 2.94 *p* < 0.01
Visceral fat [%]	23.80 ± 0.74	29.78 ± 0.89 *p* < 0.01	31.15 ± 1.95 *p* = 0.06	35.80 ± 2.96 *p* < 0.01
VAT/SAT ratio	0.30 ± 0.01	0.44 ± 0.02 *p* < 0.01	0.48 ± 0.05 *p* = 0.06	0.62 ± 0.09 *p* < 0.01
HOMA-IR index	2.38 ± 0.11	3.95 ± 0.25 *p* < 0.01	4.03 ± 0.44 *p* = 0.02	7.05 ± 1.23 *p* < 0.01

VAT/SAT ratio—visceral adipose tissue/subcutaneous adipose tissue ratio; HOMA-IR—homeostatic model assessment estimated insulin resistance; BMI—body mass index.

**Table 3 nutrients-11-00834-t003:** Descriptive statistics, odds ratios (OR), and corresponding *p*-values in the prospective study estimated using age-adjusted generalized linear models with the logit link function. CI—confidence interval.

	Non-IFG ^†^	IFG ^‡^	Odds Ratio ^§^
Δ % waist circumference (cm)	99.86 ± 0.73	103.11 ± 1.07	OR = 1.06 (CI 1.01–1.1) (*p* < 0.01)
Δ % hip circumference (cm)	98.26 ± 0.74	100.64 ± 0.91	OR = 1.05 (CI 1.0–1.0) (*p* = 0.03)
Δ % muscle mass (%)	102.77 ± 1.30	102.79 ± 1.50	OR = 0.99 (CI 0.9–1.05) (*p* = 0.98)
Δ % HOMA-IR	125.65 ± 8.42	142.80 ± 8.52	OR = 1.004 (CI 1.0–1.008) (*p* = 0.05)
Δ % visceral fat (%)	85.61 ± 4.14	97.53 ± 5.70	OR = 1.01 (CI 0.9–1.02) (*p* = 0.09)
Δ % subcutaneous fat (%)	110.51 ± 2.62	108.26 ± 3.92	OR = 0.99 (CI 0.9–1.01) (*p* = 0.62)
Δ % VAT/SAT ratio	82.34 ± 5.62	100.77 ± 9.21	OR = 1.01 (CI 0.9–1.01) (*p* = 0.09)

^†^ Non-IFG subjects who remained non-IFG at the follow-up visit; mean ± standard error; ^‡^ IFG subjects who were diagnosed with IFG at the follow-up visit; mean ± standard error; ^§^ age adjusted. VAT/SAT ratio—visceral adipose tissue/subcutaneous adipose tissue ratio; HOMA-IR—homeostatic model assessment estimated insulin resistance.

**Table 4 nutrients-11-00834-t004:** Descriptive statistics, odds ratios, and corresponding *p*-values in the prospective study estimated using age-adjusted generalized linear models with the logit link function.

	Non-IGT ^†^	IGT ^‡^	Odds Ratio ^§^
Δ % waist circumference (cm)	102.08 ± 0.69	103.03 ± 1.67	OR = 1.02 (CI 0.9–1.08) (*p* = 0.54)
Δ % hip circumference (cm)	99.34 ± 0.61	102.47 ± 0.99	OR = 1.09 (CI 1.0–1.18) (*p* = 0.04)
Δ % muscle mass (%)	102.51 ± 1.09	97.84 ± 2.92	OR = 0.93 (CI 0.8–1.02) (*p* = 0.21)
Δ % HOMA-IR	128.97 ± 6.12	139.51 ± 14.38	OR = 1.01 (CI 0.9–1.01) (*p* = 0.13)
Δ % visceral fat (%)	91.27 ± 3.95	128.66 ± 12.62	OR = 1.03 (CI 1.0–1.05) (*p* = 0.01)
Δ % subcutaneous fat (%)	110.20 ± 2.48	89.35 ± 4.77	OR = 0.92 (CI 0.8–0.97) (*p* = 0.01)
Δ % VAT/SAT ratio	90.86 ± 5.95	150.41 ± 20.22	OR =1.02 (CI 1.0–1.03) (*p* = 0.02)

^†^ Non-IGT subjects who remained non-IGT at the follow-up visit; mean ± standard error; ^‡^ IGT subjects who were diagnosed with IGT at the follow-up visit; mean ± standard error; ^§^ age adjusted.

**Table 5 nutrients-11-00834-t005:** Descriptive statistics, odds ratios, and corresponding *p*-values in the prospective study estimated using age-adjusted generalized linear models with the logit link function.

	Non-T2DM ^†^	T2DM ^‡^	Odds Ratio ^§^
Δ % waist circumference (cm)	101.93 ± 0.61	101.19 ± 2.23	OR = 0.99 (*p* = 0.91)
Δ % hip circumference (cm)	99.75 ± 0.55	97.87 ± 1.86	OR = 0.96 (*p* = 0.31)
Δ % muscle mass (%)	102.66 ± 0.96	94.02 ± 1.56	OR = 0.84 (*p* = 0.02)
Δ % HOMA-IR	127.95 ± 5.39	181.43 ± 25.92	OR = 1.01 (*p* < 0.01)
Δ % visceral fat (%)	93.89 ± 3.69	101.41 ± 5.70	OR = 1.01 (*p* = 0.41)
Δ % subcutaneous fat (%)	109.58 ± 2.66	102.30 ± 4.51	OR = 0.98 (*p* = 0.35)
Δ % VAT/SAT ratio	95.02 ± 5.60	101.72 ± 9.01	OR = 1.00 (*p* = 0.53)

^†^ T2DM not detected during the first visit; mean ± standard error; ‡T2DM detected during the follow-up visit; mean ± standard error; ^§^ age adjusted.

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
