# Peer review of "The Role of Muscle Decline in Type 2 Diabetes Development: A 5-Year Prospective Observational Cohort Study"

_nutrients, 2019, doi:10.3390/nu11040834_

Round 1

Reviewer 1 Report

This topic is unrelevant for the literature

there are so many articles related to this topic like this

Leenders, M., Verdijk, L.B., van der Hoeven, L., Adam, J.J., Van Kranenburg, J., Nilwik, R. and Van Loon, L.J., 2013. Patients with type 2 diabetes show a greater decline in muscle mass, muscle strength, and functional capacity with aging. Journal of the American Medical Directors Association14(8), pp.585-592.

Author Response

We wish to thank the Editors and Reviewers for their valuable contributions and comments.

We have submitted a revised version that addresses the comments made by the Reviewers and Editors. Our responses to each comment are detailed below.

Reviewer 1

This topic is unrelevant for the literature

there are so many articles related to this topic like this

Leenders, M., Verdijk, L.B., van der Hoeven, L., Adam, J.J., Van Kranenburg, J., Nilwik, R. and Van Loon, L.J., 2013. Patients with type 2 diabetes show a greater decline in muscle mass, muscle strength, and functional capacity with aging. Journal of the American Medical Directors Association, 14(8), pp.585-592.

Response:  We are sorry, but we can not agree with this comment. We did not evaluate the decline in muscle mass in diabetic subject, as suggested by Reviewer and provided article citation, but the aim of our study was to investigate the possible risk factors for type 2 diabetes development, in subjects who were normoglycemic at baseline, and some of them developed carbohydrate dysmetabolism during follow-up period. And there are not many papers related to a muscle decline as an independent risk factor of diabetes.

Reviewer 2 Report

In the current longitudinal prospective cohort study, the authors recruited 1,160 subjects and assessed body composition and indices of glucose tolerance and insulin sensitivity.  After 5 years they conducted follow-up assessments of these measure in an attempt reveal risk factors associated with the development of type II diabetes (T2D).  The authors report that change in muscle mass and change in HOMA-IR were predictive of progression to T2D.  Though there are some limitations and issues that are confusing (see below), this data set has the potential to contribute to the body of literature related to risk factors for T2D.

Primary Weaknesses:

·        The abstract is a bit deceiving regarding the number of subjects.  If one just reads the abstract, they would be left to believe that there were 1,160 subjects at baseline and follow-up.  In reality, only 219 of the 1,160 of the subjects were assessed at the 5 year follow-up.  That is a very low percentage at follow-up (<20%).  The authors admit that this is a limitation, and I agree.

·        I’m very confused regarding the authors use of “odds ratios”.  In health-related research odds ratios are typically used to determine the chance of some event (i.e. diagnosis of T2D) occurring in one group vs. a reference group?  In the case of this study I assume that the event is development of T2D, so what are the groups?  For instance in Table 5 the authors report and odds ratio of 0.84 for change in percent muscle mass?  How do we interpret this?  This is not the typical use of odds ratios.  Granted, I am not a statistician, but this is very confusing.

·        Also related to odds ratios, some of the ORs that they indicate as statistically significant are quite confusing.  Let’s again look at table 5.  How can an OR of 1.01 for change in % HOMA-IR possibly be significant?   If a 95% confidence interval contains 1.0, then ORs are not significant.  How can the 95% CI for an OR of 1.01 not contain “1.0”?  I merely use this table as an example, I am confused with the authors’ use of ORs throughout the paper.  Why did they not simply use linear regression?

Minor Issues:

·        The authors state that their population is middle-aged, but they then state an age range of 18-65 years on page 3, line 73.

·        The authors could better define the criteria that they used for IGT.

·        In table 1, I’m confused regarding the ratios that the authors indicate for IFG, IGT, and T2D?  The units simply indicate “N” suggesting that a single number will be presented.  It would be helpful to clarify these ratios.  I know that at least one of the numbers in the ratio is the number with a specified diagnosis.  What is the other?

·        The paper is generally well written, but there are a few wording and sentence construction issues that make reading parts of the paper difficult.  For instance, the authors frequently say “independently” when “independent” would be the correct wording (one example: page 8, line 204).  There are several other instances of this.

·        The authors also frequently use “what” inappropriately. (Examples: page 10, line 254, and page 11, line 262).

·        Lines 256 – 261 (page 11) are very confusing and need revised to for the reader.

Author Response

We wish to thank the Editors and Reviewers for their valuable contributions and comments.

We have submitted a revised version that addresses the comments made by the Reviewers and Editors. Our responses to each comment are detailed below.

Reviewer 2

In the current longitudinal prospective cohort study, the authors recruited 1,160 subjects and assessed body composition and indices of glucose tolerance and insulin sensitivity.  After 5 years they conducted follow-up assessments of these measure in an attempt reveal risk factors associated with the development of type II diabetes (T2D).  The authors report that change in muscle mass and change in HOMA-IR were predictive of progression to T2D.  Though there are some limitations and issues that are confusing (see below), this data set has the potential to contribute to the body of literature related to risk factors for T2D.

Primary Weaknesses:

·        The abstract is a bit deceiving regarding the number of subjects.  If one just reads the abstract, they would be left to believe that there were 1,160 subjects at baseline and follow-up.  In reality, only 219 of the 1,160 of the subjects were assessed at the 5 year follow-up.  That is a very low percentage at follow-up (<20%).  The authors admit that this is a limitation, and I agree.

Response:  We agree with a Reviewer that low percent of follow-up attendance is a weak point of our study and we are working on improving it in our projects.

·        I’m very confused regarding the authors use of “odds ratios”.  In health-related research odds ratios are typically used to determine the chance of some event (i.e. diagnosis of T2D) occurring in one group vs. a reference group?  In the case of this study I assume that the event is development of T2D, so what are the groups?  For instance in Table 5 the authors report and odds ratio of 0.84 for change in percent muscle mass?  How do we interpret this?  This is not the typical use of odds ratios.  Granted, I am not a statistician, but this is very confusing.

Response:  Since it was a follow-up study, patients went under examinations twice. It turned out, that with some of them the prediabetes states or type 2 diabetes development was observed, while the others remained healthy. That gave us the two desired groups for comparisons.

ODDS-RATIO = 0.84 for change in percent muscle mass (PMM) can be interpreted as follows: with constant decrease in PMM the risk of developing prediabetes/type 2 diabetes rises

For ODDS-RATIO = 1.01 for change in percent homa-ir interpretation might be: with constant increase in percent homa-ir the risk of developing prediabetes/type 2 diabetes rises

The choice of logistic regression seemed to be straightforward for the authors since the dependent variable (T2M) takes two values: 0/1.

·        Also related to odds ratios, some of the ORs that they indicate as statistically significant are quite confusing.  Let’s again look at table 5.  How can an OR of 1.01 for change in % HOMA-IR possibly be significant?   If a 95% confidence interval contains 1.0, then ORs are not significant.  How can the 95% CI for an OR of 1.01 not contain “1.0”?  I merely use this table as an example, I am confused with the authors’ use of ORs throughout the paper.  Why did they not simply use linear regression?

Response:  Odds-Ratios take values close to 1 when there is a considerable difference in scale between dependent and independent variable. Here we have % HOMA-IR of magnitude 1e+02, whereas the dependent variable (prediabetes/type 2 diabetes) takes values 0/1. Taking this further for interpretation: each additional 1% in favor of % HOMA-IR results in a 1% increase in the odds of prediabetes/type 2 diabetes. Changing scales of the independent variables would be a somehow solution – it would ease the interpretation, but would not change the results.

Minor Issues:

·        The authors state that their population is middle-aged, but they then state an age range of 18-65 years on page 3, line 73.

Response:  

We agree with a Reviewer, and we have corrected and replaced a “middle-age” with an “adults” or “adult subjects” (please see lines 21-22 page 1, line 37  page 2, line 63 page 3, line 311 page 12 ).

·        The authors could better define the criteria that they used for IGT.

Response: We defined the criteria in a better way (please see lines 109-114 page 4).

·        In table 1, I’m confused regarding the ratios that the authors indicate for IFG, IGT, and T2D?  The units simply indicate “N” suggesting that a single number will be presented.  It would be helpful to clarify these ratios.  I know that at least one of the numbers in the ratio is the number with a specified diagnosis.  What is the other?

Response:  „N” says how many subjects have prediabetes, DMt2 or were normoglicemic on first and second visits. I would like to highlight the fact that to the analysis were included patients with normoglycemia on the first visit (please see lines 83-85 page 3). We have corrected the mistake (please see Table 1). To avoid any confusions we have also added this information below the table 1 ( please see table 1).

·        The paper is generally well written, but there are a few wording and sentence construction issues that make reading parts of the paper difficult.  For instance, the authors frequently say “independently” when “independent” would be the correct wording (one example: page 8, line 204).  There are several other instances of this.

Response: We have corrected the wording (please see lines 229 page 9 ). The paper has been edited by MDPI's English editing service. I received the English Editing Certificate.

·        The authors also frequently use “what” inappropriately. (Examples: page 10, line 254, and page 11, line 262).

Response: We have corrected the wording (please see lines 279 page 10 and line 287 page 11.) The paper has been edited by MDPI's English editing service. I received the English Editing Certificate.

·        Lines 256 – 261 (page 11) are very confusing and need revised to for the reader.

Response:  We have revised and reconstructed the sentences and we hope that now they are clear (please see lines 280- 285 page 11).

Reviewer 3 Report

The manuscript by Maliszewska et al., entitled “The role of muscle decline in type 2 diabetes development: A 5-year prospective observational cohort study” aims to identify risk factors for insulin resistance and T2DM. While patient data of this nature is valuable, many aspects of the analysis are unclear and it becomes difficult to assess the validity of the conclusions.  

The study has been performed from the perspective that the measured parameters (e.g. loss of muscle mass) are risk factors for insulin resistance and T2DM, however, the possibility that insulin resistance and T2DM drive the loss of muscle mass should also be considered and discussed.

Line 22-26 in abstract. Listing the impressive numbers recruited for the initial measures but not the much smaller number of patients actually measured in the follow up is misleading. Please make it clear also in the Abstract that the conclusions of this study are not based on 1000+ subjects. Again, it should be clear that the actual number of diagnosed patients refers to the % of the 200 subjects recruited for the follow up study.

How were ‘potential risk factors’ chosen? Were these measures defined before the study was conducted?

Line 123-124, 128-130 and throughout. What was the threshold value for the correlation coefficient? There is some information as to how this was specified, but no information on actual correlation coefficients. Please include information about the strength of correlations throughout, this is just as important as whether the correlation is significant. Shouldn’t the ‘significance’ be determined by significance (i.e. a p value). The correlation rather refers to the strength of the relationship.

Line 132 – Please provide more information about how attributes were selected.

Line 154 – please report the actual correlations.

Table 3: It is unclear how these data are calculated. I guess you mean to say that all patients were non-IFG at the initial visit, and then the non-IFG group still doesn’t have IFG at the follow up but the IFG group does.

Please also include data examining the predictive value of the initial values and the follow up numbers. Is it really the delta that is important, or are the initial or follow up values the main contributor? In terms of usefulness of such a model it would seem that a real time measure is far more useful than a delta over 5 years… unless you can show that this really adds a significant improvement to the predictive power of these measures?

Line 182 – What are the 36 features and how were they chosen?

Line 189 – Please show details of the model.

Line 191-193 - “The data shows that a reduction of muscle mass increases the risk of T2DM development (OR = 0.84, p = 0.02), as well as an increase in insulin resistance estimated by the HOMA-IR calculation (OR = 1.01, p < 0.01).” It is unclear what data the second sentence refers to.  

Table 5, delta % HOMA-IR – It is unclear how there can be a statistical difference and an odds ratio of almost exactly 1 (i.e. no change in risk). In fact, all odds ratio calculations seem strange – on line 192 the odds ratio is 0.84 (which should be a lower risk) but then you state that a reduction in muscle mass increases the risk of T2DM. Please provide more detail as to how these values were calculated.  

English is not bad, but it would improve readability to have a native speaker edit the manuscript.

Reviewer 4 Report

 The authors evaluated the possible risk factors associated with Type 2 diabetes (T2DM) development in middle-aged subjects, during a 5-year prospective observational cohort study. The topic is of interest and the manuscript is well written. Although the number of patients in each event (IFG, IGT, T2D) is small, the findings are very interesting.

1)   The authors use sarcopenia as loss of skeletal muscle mass. However, the definition of sarcopenia is loss of skeletal muscle mass with decreased muscle strength or physical activity. So, they should correctly use loss of skeletal muscle mass instead of sarcopenia.

2)   This is a single center study performed in Poland. Therefore, external validation in other countries or races is necessary to confirm the findings of this study.

3)   There is no description regarding the definition of IFG and IGT. Please show these definitions in Materials and Methods section.

4)   Please add approved number of the Ethical Committee.

5)   In the abstract, the authors should spell out T2DM in the first appearance.

Author Response

We wish to thank the Editors and Reviewers for their valuable contributions and comments.

We have submitted a revised version that addresses the comments made by the Reviewers and Editors. Our responses to each comment are detailed below.

Reviewer 4

The authors evaluated the possible risk factors associated with Type 2 diabetes (T2DM) development in middle-aged subjects, during a 5-year prospective observational cohort study. The topic is of interest and the manuscript is well written. Although the number of patients in each event (IFG, IGT, T2D) is small, the findings are very interesting.

Response:  We wish to thank the Reviewer for this comment.

1)   The authors use sarcopenia as loss of skeletal muscle mass. However, the definition of sarcopenia is loss of skeletal muscle mass with decreased muscle strength or physical activity. So, they should correctly use loss of skeletal muscle mass instead of sarcopenia.

Response:  I have corrected this ( please see line 249 page 10). In another parts of our manuscript I used the word “sarcopenia” when I quoted another authors’ papers (please see lines 269 page 11 refers to reference No 2; line 54 page 2 and line 246 page 10 regards references No 3 and 5)

2)   This is a single center study performed in Poland. Therefore, external validation in other countries or races is necessary to confirm the findings of this study.

Response:  We agree, and we have underlined it (please see lines 305 to 307 page 11).

3)   There is no description regarding the definition of IFG and IGT. Please show these definitions in Materials and Methods section.

Response: We described the IFG and IGT in text (please see lines 109-114 page 4).

4)   Please add approved number of the Ethical Committee.

Response:

We have added the number of the Ethical Committee (please see lines 72 page 3).

5)   In the abstract, the authors should spell out T2DM in the first appearance.

Response:   We described T2DM type 2 diabetes mellitus  (please see line 20 page 1).

Round 2

Reviewer 1 Report

this manuscript need to be revised in order to be accepted.

-add CI 95% on odds ratio in table 3 and 4

-add in table 1 the history of T2D ( mean of years)

-table 4, revise the p value in according table 3 and 2. please state p> when appropriated

- in table 2 is reported muscle mass. I think that is better clarify this point because probably free fat mass is to be reported.

- delta change must be correlated to demonstrate the possible associations.

Author Response

We wish to thank the Editors and Reviewers for their valuable contributions and comments.

Our manuscript has been significantly improved by incorporating their suggestions.

We have submitted a revised version that addresses the comments made by the Reviewers and Editors. Our responses to each comment are detailed below.

Reviewer 1

Reviewer 1:  Add CI 95% on odds ratio in table 3 and 4

Response:  We added the CI 95% to the tables ( please see Table 3 and Table 4).

Reviewer 1: Add in table 1 the history of T2D ( mean of years)

Response: Subjects did not have any history of T2DM (please see line 81). The T2DM were recognized for the first time based on results from the visit.

Reviewer 1: table 4, revise the p value in according table 3 and 2. please state p> when appropriated

Response: Thank you for this comment. I had formatted it (please see the Table 4).

Reviewer 1: in table 2 is reported muscle mass. I think that is better clarify this point because probably free fat mass is to be reported.

Response: In the Table 2 we reported the mean value of muscle mass (not fat free mass), estimated in % in each group of subjects (please see Table 2).

Reviewer 1: delta change must be correlated to demonstrate the possible associations.

Response: The lowest sample estimate for correlation coefficient (when checking the before and after variables) was equal to 0.50 (Spearman's correlation coefficient) and corresponding p-value was less than 2.2e-16.

Reviewer 2 Report

I still have issues with the paper that the authors have not adequately resolved in their revision.

·        The authors have still not clarified in the number of subjects that underwent follow-up in their abstract.  That needs to be corrected.  If a reader just views the abstract (as many readers do), they will be misled to believe that 1,160 subjects underwent baseline and follow-up testing.

·        The use odds ratios (and the stated significance for some of the ORs), still do not make sense.  I don’t know what to offer beyond that.  How an odds ratio of 1.01 could possibly be significant is beyond me.  At a minimum, the authors need to back that up by showing the 95% CI’s.  If the 95% CI’s contain “1.0” in the range they would not be significant.  It seems improbable the 95% CI for an OR of 1.01 would not contain 1.0.  I am not the only reviewer that expressed this concern.  The authors' response to this concern does not make sense to me.

Author Response

We wish to thank the Editors and Reviewers for their valuable contributions and comments.

Our manuscript has been significantly improved by incorporating their suggestions.

We have submitted a revised version that addresses the comments made by the Reviewers and Editors. Our responses to each comment are detailed below.

Reviewer 2

I still have issues with the paper that the authors have not adequately resolved in their revision.

·        The authors have still not clarified in the number of subjects that underwent follow-up in their abstract.  That needs to be corrected.  If a reader just views the abstract (as many readers do), they will be misled to believe that 1,160 subjects underwent baseline and follow-up testing.

Response: We corrected (please see line 25-26 page 1).

·        The use odds ratios (and the stated significance for some of the ORs), still do not make sense.  I don’t know what to offer beyond that.  How an odds ratio of 1.01 could possibly be significant is beyond me.  At a minimum, the authors need to back that up by showing the 95% CI’s.  If the 95% CI’s contain “1.0” in the range they would not be significant.  It seems improbable the 95% CI for an OR of 1.01 would not contain 1.0.  I am not the only reviewer that expressed this concern.  The authors' response to this concern does not make sense to me.

Response: As it is presented in an updated manuscript – none of the confidence intervals for the “significant” OR contain 1.

Referring delta % Homa-IR OR. It comes out of the definition: if the value of the independent variable changes by 1 – our variable is a percent, so it is “quite common” to have this kind of changes, nevertheless significant, the risk = OR*RISK_0, where RISK_0 is the baseline risk for developing T2DM. For example: if we had OR=1.4, then change in delta % Homa-IR=4 would involve the risk=1.4*1.4*1.4*1.4=3.8416, which is huge. This “misunderstanding” comes out of the nature of variables: some of them take values from 1-3 and some of them take values from 20-40. It is straightforward then, that when dealing with variables with small variability range one might suspect considerably higher or considerably lower than 1 values of odds ratio. Therefore, variables like % change in HOMA-IR having wide range of values, are accompanied by relatively small odds ratios, but still significant.

Reviewer 3 Report

The authors have answered some of my questions, but many issues remain that require clarification or the addition of further details.

Comment Round 1: The study has been performed from the perspective that the measured parameters (e.g. loss of muscle mass) are risk factors for insulin resistance and T2DM, however, the possibility that insulin resistance and T2DM drive the loss of muscle mass should also be considered and discussed.

Response:  We have added in the Discussion section (please see lines 269 - 285 page 11 ).

Comment Round 2: Page 11 – 280-283: The discussion added in response to my previous comment does not make sense. For example: One of the potential mechanisms mediating the development of sarcopenia and insulin resistance is reduced expression of the E3 ubiqitin enzymes, which participate in muscle protein degradation, a mechanism not observed in the age-related sarcopenia [27]“.

Firstly, sarcopenia is the age-related loss of muscle mass and doesn’t relate to other forms of muscle loss. Secondly, the reference (#27) does not support the claim.

Please focus on addressing the comment. As you are only reporting correlations, you have no way of establishing cause and consequence when comparing a change over time (i.e. change in muscle mass) with a change from non-T2DM to T2DM.  

Comment Round 1: Line 22-26 in abstract. Listing the impressive numbers recruited for the initial measures but not the much smaller number of patients actually measured in the follow up is misleading. Please make it clear also in the Abstract that the conclusions of this study are not based on 1000+ subjects. Again, it should be clear that the actual number of diagnosed patients refers to the % of the 200 subjects recruited for the follow up study.

Response:  We added this information please see line 34 and 35 page 2.

Comment round 2: Please move this information to line 24-25 when the follow up visit is first mentioned.

Comment round 1: Line 123-124, 128-130 and throughout. What was the threshold value for the correlation coefficient? There is some information as to how this was specified, but no information on actual correlation coefficients. Please include information about the strength of correlations throughout, this is just as important as whether the correlation is significant. Shouldn’t the ‘significance’ be determined by significance (i.e. a p value). The correlation rather refers to the strength of the relationship.

Response:  For every feature under consideration, a Spearman’s correlation coefficient was calculated in order to eliminate redundancy – there is no need for the inclusion of correlated features – it complicates the model aiming at T2D prediction. The applied procedure was as follows: (1) take the significant features (selected in the Geleralized Linear Models step) and calculate the Spearman’s correlation coefficient for every pair of features, (2) due to the multiple testing problem – apply FDR p-value correction procedure to the list of p-values obtained in the previous step, (3) select the highest value of correlation coefficient, when the overall correlation is not significant, c.1, and the lowest significant correlation, c.2, (4) with the use of the findCorrelation function from the caret package and a correlation threshold value (between c.1 = 0.21 and c.2 = 0.29 and set = 0.29) a correlation matrix was searched and a vector of integers corresponding to columns to remove to reduce pair-wise correlations returned. Pointed features were excluded from classification procedure.

When referring ‘significant’ – p-value < 0.05

Comment round 2: I disagree that it is not important to report the Spearman’s correlation coefficient. You report whether the relationship is significant and what the odds ratio is… but not how strong of a relationship the odds ratio calculation is based on.

Also, please add any explanations provided here that are not already included in the manuscript.

Comment round 1: Line 154 – please report the actual correlations.

Response:  

Due to the fact, that the overall number of correlations is quite big (each pair of 36) we can provide a file containing requested information. We attached the file in pdf.

Comment round 2: Please add the correlations to any table reporting an odds ratio. There is no need to present correlations where you have not presented an odds ratio. An attached PDF is not sufficient.

Comment 1: Line 182 – What are the 36 features and how were they chosen?

Response:  The 36 features are listed below:

1.‘BMI change ”               

2. BMI percentage change”

3. WHI change”                  

4. WHI percentage change”

5. waist circumference percentage change             

 6. waist circumference change“

7. hip circumferance percentage change                      

  8. hip circumferance change

9.percatnage change of fasting glucose   

 10. change of fasting glucose

11.HOMA-IR percentage change  

12. HOMA_IR change”

13.HOMA-B percentage change     

 14. HOMA_B change”

15.HbA1c percentage change   

16. HbA1c change”

17. Change of  Fat mass (%)

18.Percentage change of fat mass (%)

19. Percentage change of muscle mass (kg)

20. Change of muscle mass (kg)

21. Percentage Change of VAT(cm³)

22.Change of SAT (cm³)

23. Percentage change of SAT(%)

24. change of muscle mass (%)

25. Percentage change of muscle mass (%)

26. Change of VAT (%)

27. Percnetage change of VAT (%)

28. percentage change of SAT (cm³)

29. Change of SAT (%)

30. Change of  VAT/SAT ratio

31. Percentage change of VAT/SAT ratio

32. Change of glucose (in 2h of OGTT)

33.Percenatge change of glucose (in 2h of OGTT)

34. Change of fat mass (kg)

35. Percentage change of fat mass (%)

36. Change of VAT (cm³)We chose listed above features, which in our opinion may be associated with type 2 diabetes development.

Comment round 2: Please include these in the manuscript. It is also helpful for future researchers to be able to check which features do not significantly impact T2DM development.

Comment round 1: Line 189 – Please show details of the model.

Response:  Thirty six models were constructed, each consisting patient’s age as a confounding factor and a variable to be checked – each variable of 36 was included separately in its model. If a particular model showed to be significant – it terms of this additional variable (not age) - then this variable was remained for further consideration.

Comment round 2: As a minimum, please include the equation that could be used to predict T2DM development. In the abstract you state that HOMA-IR and loss of muscle mass combined had the best classification accuracy.  

Comment round 1: Table 5, delta % HOMA-IR – It is unclear how there can be a statistical difference and an odds ratio of almost exactly 1 (i.e. no change in risk). In fact, all odds ratio calculations seem strange – on line 192 the odds ratio is 0.84 (which should be a lower risk) but then you state that a reduction in muscle mass increases the risk of T2DM. Please provide more detail as to how these values were calculated. 

Response:  Referring the OR = 0.84. Following the definition of OR: if the value of the independent variable (muscle mass) rises by 1, then the risk is 0.84*RISK_0, where RISK_0 is the baseline risk for developing T2DM. Then, when we look from another perspective: if the value of the independent variable falls (decline in muscle mass), then the risk rises.

Referring delta % Homa-IR OR. It comes out of the definition: if the value of the independent variable changes by 1 – our variable is a percent, so it is “quite common” to have this kind of changes, nevertheless significant. For example: if we had OR=1.4, then change in delta % Homa-IR=4 would involve the risk=1.4*1.4*1.4*1.4=3.8416, which is huge.

Comment round 2: Okay, I understand now. You are using values from your regression analysis to calculate an odds ratio rather than the traditional (or simple) way of separating the independent variable into groups (i.e. loss of muscle mass >5% and <5%) and then calculating the change in risk of developing T2DM in the group with more muscle loss.

This needs to be properly explained in the statistical methods section and it would be helpful to express these values in a logical way. For muscle mass, this may be better to use %loss of muscle mass, rather than %muscle mass… as you later talk about muscle loss as increasing risk. It may also be more helpful to choose appropriate units and clearly outline them in the tables… otherwise it can be unclear as to the impact the increased risk is associated with. For HOMA-IR, an increased risk of 1.01 appears very low, but when you consider there is a rather large change in the units then it explains the small odds ratio.

Author Response

We wish to thank the Editors and Reviewers for their valuable contributions and comments. Our manuscript has been significantly improved by incorporating their suggestions.

We have submitted a revised version that addresses the comments made by the Reviewers and Editors. Our responses to each comment are detailed below.

Reviewer 3

The authors have answered some of my questions, but many issues remain that require clarification or the addition of further details.

Comment Round 1: The study has been performed from the perspective that the measured parameters (e.g. loss of muscle mass) are risk factors for insulin resistance and T2DM, however, the possibility that insulin resistance and T2DM drive the loss of muscle mass should also be considered and discussed.

Response:  We have added in the Discussion section (please see lines 269 - 285 page 11 ).

Comment Round 2: Page 11 – 280-283: The discussion added in response to my previous comment does not make sense. For example: “One of the potential mechanisms mediating the development of sarcopenia and insulin resistance is reduced expression of the E3 ubiqitin enzymes, which participate in muscle protein degradation, a mechanism not observed in the age-related sarcopenia [27]“.

Firstly, sarcopenia is the age-related loss of muscle mass and doesn’t relate to other forms of muscle loss. Secondly, the reference (#27) does not support the claim.

Response Round 2: Thank you for this comment. We have edited it (please see line 297-301 page 12). We are sorry, but we can not agree with a comment that the references (#27) does not support the claim- (please seen in the paper by Cleasby ME, Jamieson PM, Atherton PJ Insulin resistance and sarcopenia: mechanistic links between common co-morbidities. J Endocrinol. 2016 May;229(2):R67-81; the explanation is on page R72  paragraph – Other molecular pathways potentially mediating the development of both sarcopenia and IR).

Please focus on addressing the comment. As you are only reporting correlations, you have no way of establishing cause and consequence when comparing a change over time (i.e. change in muscle mass) with a change from non-T2DM to T2DM.  

Response: We corrected it (please line 37-38 page 2, and 311 -312 page 12).

Comment Round 1: Line 22-26 in abstract. Listing the impressive numbers recruited for the initial measures but not the much smaller number of patients actually measured in the follow up is misleading. Please make it clear also in the Abstract that the conclusions of this study are not based on 1000+ subjects. Again, it should be clear that the actual number of diagnosed patients refers to the % of the 200 subjects recruited for the follow up study.

Response:  We added this information please see line 34 and 35 page 2.

Comment round 2: Please move this information to line 24-25 when the follow up visit is first mentioned.

Response round 2: We edited it as suggested (please see line 25-26).

Comment round 1: Line 123-124, 128-130 and throughout. What was the threshold value for the correlation coefficient? There is some information as to how this was specified, but no information on actual correlation coefficients. Please include information about the strength of correlations throughout, this is just as important as whether the correlation is significant. Shouldn’t the ‘significance’ be determined by significance (i.e. a p value). The correlation rather refers to the strength of the relationship.

Response round 2:  For every feature under consideration, a Spearman’s correlation coefficient was calculated in order to eliminate redundancy – there is no need for the inclusion of correlated features – it complicates the model aiming at T2D prediction. The applied procedure was as follows: (1) take the significant features (selected in the Geleralized Linear Models step) and calculate the Spearman’s correlation coefficient for every pair of features, (2) due to the multiple testing problem – apply FDR p-value correction procedure to the list of p-values obtained in the previous step, (3) select the highest value of correlation coefficient, when the overall correlation is not significant, c.1, and the lowest significant correlation, c.2, (4) with the use of the find Correlation function from the caret package and a correlation threshold value (between c.1 = 0.21 and c.2 = 0.29 and set = 0.29) a correlation matrix was searched and a vector of integers corresponding to columns to remove to reduce pair-wise correlations returned. Pointed features were excluded from classification procedure.

When referring ‘significant’ – p-value < 0.05

Comment round 2: I disagree that it is not important to report the Spearman’s correlation coefficient. You report whether the relationship is significant and what the odds ratio is… but not how strong of a relationship the odds ratio calculation is based on.

Response round 2: The value of odds ratio comes directly from the model, which has nothing in common with correlation coefficients.

Example model: T2M ~ HOMA.IR.change + age

Where variables on the right hand side are independent variables (in terms of modelling). We are seeking an explanation how changes in HOMA.IR influence the risk of developing T2DM.

To obtain odds ratio one needs to power e (constant) to the power of an appropriate coefficient obtained after model fitting.  

Comment: Also, please add any explanations provided here that are not already included in the manuscript.

Response round 2: Statistical method has been up-dated (please see lines 127-163 in manuscript 2.6 Statistical analysis ).

Comment round 1: Line 154 – please report the actual correlations.

Response: Due to the fact, that the overall number of correlations is quite big (each pair of 36) we can provide a file containing requested information. We attached the file in pdf.

Comment round 2: Please add the correlations to any table reporting an odds ratio. There is no need to present correlations where you have not presented an odds ratio. An attached PDF is not sufficient.

Response round 2: As explained above – there is no relationship between Spearman’s correlations and odds ratios.

Comment 1: Line 182 – What are the 36 features and how were they chosen?

Response:  The 36 features are listed below:

1.‘BMI change ”               

2. BMI percentage change”

3. WHI change”                  

4. WHI percentage change”

5. waist circumference percentage change             

 6. waist circumference change“

7. hip circumferance percentage change                      

  8. hip circumferance change

9.percatnage change of fasting glucose   

 10. change of fasting glucose

11.HOMA-IR percentage change  

12. HOMA_IR change”

13.HOMA-B percentage change     

 14. HOMA_B change”

15.HbA1c percentage change   

16. HbA1c change”

17. Change of  Fat mass (%)

18.Percentage change of fat mass (%)

19. Percentage change of muscle mass (kg)

20. Change of muscle mass (kg)

21. Percentage Change of VAT(cm³)

22.Change of SAT (cm³)

23. Percentage change of SAT(%)

24. change of muscle mass (%)

25. Percentage change of muscle mass (%)

26. Change of VAT (%)

27. Percnetage change of VAT (%)

28. percentage change of SAT (cm³)

29. Change of SAT (%)

30. Change of  VAT/SAT ratio

31. Percentage change of VAT/SAT ratio

32. Change of glucose (in 2h of OGTT)

33.Percenatge change of glucose (in 2h of OGTT)

34. Change of fat mass (kg)

35. Percentage change of fat mass (%)

36. Change of VAT (cm³)

We chose listed above features, which in our opinion may be associated with type 2 diabetes development.

Comment round 2: Please include these in the manuscript. It is also helpful for future researchers to be able to check which features do not significantly impact T2DM development.

Response round 2: We added it to the statistical methods description (please see line 134-148 page 5-6).

Comment round 1: Line 189 – Please show details of the model.

Response:  Thirty six models were constructed, each consisting patient’s age as a confounding factor and a variable to be checked – each variable of 36 was included separately in its model. If a particular model showed to be significant – it terms of this additional variable (not age) - then this variable was remained for further consideration.

Comment round 2: As a minimum, please include the equation that could be used to predict T2DM development. In the abstract you state that HOMA-IR and loss of muscle mass combined had the best classification accuracy.  

Response round 2: Unfortunately, there does not exist a simple equation, which enables classification, e.g.: x + 2y > 3, then T2DM. Support Vector Machines classification technique is based on optimal hyperplanes separating classes, which are defined by support vectors. The number of these support vectors might be considerably big: more than 50 in our case. The knowledge, that HOMA-IR and loss of muscle mass provide good classification accuracy might be used for future research in the following way: having a set of data divide it into a training and a test set using HOMA-IR and decline in muscle mass as features; train the SVM classifier on the training data – build the classifier (extract support vectors etc. – there does not exist one perfect SVM classifier for all data of the type we have) and classify the remaining objects using the classifier that was just built.

Comment round 1: Table 5, delta % HOMA-IR – It is unclear how there can be a statistical difference and an odds ratio of almost exactly 1 (i.e. no change in risk). In fact, all odds ratio calculations seem strange – on line 192 the odds ratio is 0.84 (which should be a lower risk) but then you state that a reduction in muscle mass increases the risk of T2DM. Please provide more detail as to how these values were calculated. 

Response:  Referring the OR = 0.84. Following the definition of OR: if the value of the independent variable (muscle mass) rises by 1, then the risk is 0.84*RISK_0, where RISK_0 is the baseline risk for developing T2DM. Then, when we look from another perspective: if the value of the independent variable falls (decline in muscle mass), then the risk rises.

Referring delta % Homa-IR OR. It comes out of the definition: if the value of the independent variable changes by 1 – our variable is a percent, so it is “quite common” to have this kind of changes, nevertheless significant. For example: if we had OR=1.4, then change in delta % Homa-IR=4 would involve the risk=1.4*1.4*1.4*1.4=3.8416, which is huge.

Comment round 2: Okay, I understand now. You are using values from your regression analysis to calculate an odds ratio rather than the traditional (or simple) way of separating the independent variable into groups (i.e. loss of muscle mass >5% and <5%) and then calculating the change in risk of developing T2DM in the group with more muscle loss.

This needs to be properly explained in the statistical methods section and it would be helpful to express these values in a logical way. For muscle mass, this may be better to use %loss of muscle mass, rather than %muscle mass… as you later talk about muscle loss as increasing risk. It may also be more helpful to choose appropriate units and clearly outline them in the tables… otherwise it can be unclear as to the impact the increased risk is associated with. For HOMA-IR, an increased risk of 1.01 appears very low, but when you consider there is a rather large change in the units then it explains the small odds ratio.

Response round 2: We replaced it with a % loss of muscle mass, according to suggestions (please see line 30 and 33 page 1, line 37-38 page 2, line 132 and 142-143 page 5, line 223 page 9, line 226 page 9, line 227-228 page 10, line 230 page 10,  line 249 page 10, line 325 page 13).

Round 3

Reviewer 2 Report

I still find the authors' use of odds ratios confusing, and I fear that this will confuse readers as well.  The authors' response did not clarify or alleviate this concern for me.  That said, I fully acknowledged that I am not a statistician, and that the authors' use may be correct.  I just worry that many others will be confused as well.